# Survival, Growth Performance, and Hepatic Antioxidant and Lipid Profiles in Infected Rainbow Trout (*Oncorhynchus mykiss*) Fed a Diet Supplemented with Dihydroquercetin and Arabinogalactan

**DOI:** 10.3390/ani13081345

**Published:** 2023-04-14

**Authors:** Irina V. Sukhovskaya, Liudmila A. Lysenko, Natalia N. Fokina, Nadezhda P. Kantserova, Ekaterina V. Borvinskaya

**Affiliations:** 1Laboratory of Environmental Biochemistry, Institute of Biology, Karelian Research Centre of the Russian Academy of Sciences, 185910 Petrozavodsk, Russia; sukhovskaya@inbox.ru (I.V.S.); l-lysenko@yandex.ru (L.A.L.); fokinann@gmail.com (N.N.F.); 2Institute of Biology, Irkutsk State University, 664025 Irkutsk, Russia; borvinska@gmail.com

**Keywords:** rainbow trout, dihydroquercetin, arabinogalactan, bacterial infection, antioxidant system, lipids

## Abstract

**Simple Summary:**

Cultivated fish face multiple stressors that impact their viability, growth, and health; additionally, environmental stressors provoke the transmission of bacterial diseases in fish populations. It is widely recognized that feed supplements of plant origin can alleviate stress in fish via the stimulation of non-specific defense responses. We tested the effects of a dietary mix consisting of an antioxidant, dihydroquercetin, and a prebiotic, arabinogalactan, on growth and biochemical indices in farmed rainbow trout, *Oncorhynchus mykiss*. The supplement has been shown to maintain the viability of fish affected by a natural bacterial infection as well as alter the fatty acid composition and decrease oxidative damage to the liver. Thus, plant-origin substances could be readily available and safe alternatives to pharmacotherapy and used as a way to improve the health of fish and promote their ability to tolerate stressors under intensive production.

**Abstract:**

Natural feed supplements have been shown to improve fish viability, health, and growth, and the ability to withstand multiple stressors related to intensive cultivation. We assumed that a dietary mix of plant-origin substances, such as dihydroquercetin, a flavonoid with antioxidative, anti-inflammatory, and antimicrobial properties, and arabinogalactan, a polysaccharide with immunomodulating activity, would promote fish stress resistance and expected it to have a protective effect against infectious diseases. Farmed rainbow trout fish, *Oncorhynchus mykiss*, received either a standard diet or a diet supplemented with 25 mg/kg of dihydroquercetin and 50 mg/kg of arabinogalactan during a feeding season, from June to November. The fish in the control and experimental groups were sampled twice a month (eight samplings in total) for growth variable estimations and tissue sampling. The hepatic antioxidant status was assessed via the quantification of molecular antioxidants, such as reduced glutathione and alpha-tocopherol rates, as well as the enzyme activity rates of peroxidase, catalase, and glutathione-S-transferase. The lipid and fatty acid compositions of the feed and fish liver were analyzed using thin-layer and high-performance liquid chromatography. The viability, size, and biochemical indices of the fish responded to the growth physiology, environmental variables such as the dissolved oxygen content and water temperature, and sporadic factors. Due to an outbreak of a natural bacterial infection in the fish stock followed by antibiotic treatment, a higher mortality rate was observed in the fish that received a standard diet compared to those fed supplemented feed. In the postinfection period, reduced dietary 18:2n-6 and 18:3n-3 fatty acid assimilation contents were detected in the fish that received the standard diet in contrast to the supplemented diet. By the end of the feeding season, an impaired antioxidant response, including reduced glutathione S-transferase activity and glutathione content, and a shift in the composition of membrane lipids, such as sterols, 18:1n-7 fatty acid, and phospholipids, were also revealed in fish fed the standard diet. Dietary supplementation with plant-origin substances, such as dihydroquercetin and arabinogalactan, decreases lethality in fish stocks, presumably though the stimulation of natural resistance in farmed fish, thereby increasing the economic efficacy during fish production. From the sustainable aquaculture perspective, natural additives also diminish the anthropogenic transformation of aquaculture-bearing water bodies and their ecosystems.

## 1. Introduction

Rainbow trout, *Oncorhynchus mykiss*, is one of the most cage-cultured fish species in high-latitude countries due to its fast growth in low-temperature environments [1,2]. Unlike wild conspecifics, intensively farmed fish are known for having an altered physiological state and increased susceptibility to infections [3]. In order to sustain farm ecosystems and improve fish welfare under environmental or human-derived stressors, a number of biologically active compounds is proposed to be introduced into the diet of farmed fish [4,5,6]. Some biologically active substances of plant origin, such as dihydroquercetin (also known as taxifolin) and arabinogalactan, are predicted to promote fish stress resistance and reduce the need for antimicrobial treatments under intensive cultivation. Dihydroquercetin is a natural polyphenol that is extracted from larch (*Larix gmelinii* and *L. sibirica*) and is structurally similar to other bioactive flavonoids such as quercetin, hesperidin, or rutin. It is widely used in the medical and food industries due to its antioxidative, anti-inflammatory, antimicrobial, and antitumor activities [7,8,9,10]. There have also been attempts to use dietary dihydroquercetin in veterinary medicine, which have had either beneficial effects by increasing the immune status of gilthead seabream [11] and suppressing Cd toxicity in zebrafish embryos [12] or no impact on growth performance or any of the studied physiological variables [13]. Arabinogalactan, a plant polysaccharide, was reported to have immunomodulator activity through protecting symbiont intestinal microbiota [14,15,16,17]. Because dihydroquercetin was established to have strong antimicrobial effects in vivo [18,19,20] and arabinogalactan was reported to eliminate bacterial pathogens [15,21,22,23,24], their combination is expected to have a protective effect against infectious diseases. Both substances are promoted by their producers to fish farms as feed supplements; however, the experimental data on their benefits in the fish-rearing industry are limited [11,25,26,27,28].

The aim of the study was to estimate some physiological and biochemical indices in farmed rainbow trout (*Oncorhynchus mykiss*) that received either a standard diet or a diet supplemented with dihydroquercetin and arabinogalactan throughout a growing season for five months. Due to an episode of a natural bacterial infection that occurred during the observation period, the antioxidant and immunomodulating properties of a natural additive were able to be verified.

## 2. Materials and Methods

### 2.1. Fish Maintenance and Feeding

The experiment was carried out from June to November on a trout aquafarm in Ladmozero Lake in Northwest Russia. Healthy rainbow trout (*Oncorhynchus mykiss*) yearlings (average weight 100.1 ± 2.3 g, age 1+) were placed in four cages with an initial stocking density of 2.1 kg/m^3^; the total biomass of the fish in each cage was around 886 kg. The fish were fed one of two diets (in duplicate), either the commercial diet BioMar (Denmark; for composition, see Appendix A) without any supplements (control diet) or the same commercial diet supplemented with 25 mg/kg of dihydroquercetin and 50 mg/kg of arabinogalactan in accordance with the producer’s recommendations (quality and safety certificate no. 396-08.17, Ametis, Russia). The feed was delivered to the fish manually twice a day. The amount of food rations was calculated daily by considering the current biomass in each cage, water temperature, and dissolved oxygen content. Food fractions (3.0 or 4.5 mm) were chosen based on fish size; the compositions of both fractions of feed were similar (Appendix A). The lipid composition of the commercial feed pellets was analyzed and the results are presented in Appendix A. The supplement (dihydroquercetin and arabinogalactan) was top-dressed as feed granules by the staff of the farm directly on the day of feeding. For this, a portion of the supplement was dissolved in water at 50 °C in a 10 L tank and the solution was sprayed onto feed pellets while mixing the feed manually. During the five-month experiment, five courses of supplementation lasting for 10 to 14 days (shown in pink in the figures) were given. Feeding the supplement in short, repeated courses made it possible to achieve the desired biological effect at reduced economic cost.

The water temperature and dissolved oxygen content values measured daily with S9 Seven2Go pro (Mettler Toledo, Switzerland) throughout the study period were between 6 and 18 °C and 5 and 11 mg/L, respectively (Figure 1). The water quality variables, both directly in the cages and within 500 m from the fish farm, were assessed twice during the observation period on 29 June and 4 October (Appendix A), and all key parameters were found to be satisfactory.

During the observation period, a sporadic bacterial disease occurred in the fish stock, with the initial manifestation on 12 July in rainbow trout yearlings in both groups. The infected fish swam on their sides, refused food, and had hepatic abnormalities and hemorrhages in the liver. The infection was identified as a bacterial hemorrhagic septicemia-type disease caused by an association of *Pseudomonas putida* and *Cytophaga psychrophile.* Two weeks after the disease manifestations, the fish in all cages were treated with an antibiotic, enrofloxacin (25 mg/kg of fish weight), using a six-day regimen.

### 2.2. Fish Sampling

At the indicated sampling dates, eight fish from each cage were euthanized with an overdose of clove oil (250 mg L⁻^1^). The liver samples were dissected with scissors, then frozen in liquid nitrogen and stored at −80 °C until the enzymatic assay and evaluation of reduced glutathione and vitamin contents. For the lipid analysis, hepatic samples were fixed in 97% ethanol (with the addition of 0.001% butylated oxytoluene as an antioxidant) and stored at +4 °C.

### 2.3. Biometric Indices and Mortality

Fish mortality data were provided by the fish farm staff as the number of fish deaths on the water surface in the cages per month. Twice during the observation period (26 July, 16 November), the cages were lifted from the depths to the count dead fish accumulated at the bottom of the cages. Thus, the November mortality values included the number of unnoticed dead fish accumulated at the bottom of the cages since August (Appendix A). The total wet mass of the fish was measured using a portable Ohaus Scout SPX Series balance instrument (Figure 2).

### 2.4. Biochemical Assay

Biochemical analyses of hepatic antioxidants and lipid profile in the rainbow trout *O. mykiss* were conducted in the Equipment Sharing Center of the Karelian Research Center of the Russian Academy of Sciences (Petrozavodsk, Russia).

#### 2.4.1. Antioxidant Analyses

The frozen 0.1–0.4 g liver samples were homogenized with a Digital Disruptor Genie unit (Scientific Industries, Bohemia, NY, USA) in 5.0 mmol Tris-HCl buffer, pH 7.5. The homogenate was centrifuged at 60,000× *g* for 1 h at 4 °C in an Allegra 64R centrifuge (Beckman Coulter, Brea, USA). The resultant supernatant was used for a biochemical assay of the enzyme activities and GSH level.

The glutathione S-transferase (GST) activity was determined from the rate of reduced glutathione (GSH) binding with the 1-chloro-2,4-dinitrobenzene (CDNB) substrate [29]. A microplate well was injected with 0.225 µL of reaction mixture containing 1 mmol of CDNB and 1 mmol of GSH in 0.125 M of PBS at pH 6.5. The reaction was initiated by adding 0.025 µL of the homogenate solution, and the following increase in the solution’s optical density was recorded at 340 nm at 25 °C with a CLARIOstar microplate reader (BMG Labtech, Ortenberg, Germany). The relative activity of the enzyme in the fish tissue was expressed in µmol of the reaction product generated within a minute, calculated per mg of soluble protein in the tissue (µmol/mg protein/min).

The catalase (CAT) activity was determined using a technique by Beers and Sizer [30]. The activity of the enzyme was measured by decomposing 25 mmol of hydrogen peroxide in 50 mmol of PBS at pH 7.4 and 25 °C. The optical density of the resulting solution at 240 nm was registered. The relative CAT activity was expressed in hydrogen peroxide µmol decomposed within a minute and calculated per mg of soluble protein in the tissue, and is reported in µmol/mg protein/min.

The peroxidase (Px) activity was determined using a technique used by Maehly and Chance [31] in a reaction mixture containing 10 mmol of guaiacol and 25 mmol of hydrogen peroxide in 50 mmol of PBS at pH 7.4 and 25 °C. The peroxidase activity was determined by measuring the absorbance of the reaction product at 470 nm and is reported in µmol/mg protein/min.

The concentration of reduced glutathione (GSH) was determined based on the Cohn and Lyle method [32] modified by Hissin and Hilf [33] as described below. Briefly, soluble proteins were precipitated from the homogenate using 5% trichloroacetic acid and separated by centrifuging at 2500× *g* for 15 min. The supernatant was adjusted to pH 8.5 with 5 mol of NaOH and then mixed with 0.4 mol of Tris-HCl at pH 8.5, containing 5 mmol of ethylene diamine tetraacetic acid (EDTA). Then, 0.01% ortho-phthalaldehyde in methanol, prepared immediately before use, was added to the reaction mixture. The mixture was stirred and incubated at room temperature for 15 min, after which its fluorescence was measured (em 420 nm, ex 350 nm wavelengths). The reduced glutathione concentration was calculated according to the calibration curve plotted using series of solutions with different GSH concentrations in 0.4 mol of Tris-HCl at pH 8.5, containing 5 of mmol EDTA. The relative glutathione concentration was expressed in GSH µg per mg protein.

The soluble protein concentration in the supernatant was measured spectrophotometrically through peptide bond absorbance at 220 nm at 26 °C [34].

The concentration of alpha-tocopherol (vitamin E) in the fish liver was determined using high-performance liquid chromatography (HPLC). The samples of tissues were homogenized in 0.25 mol of sucrose solution at pH 7.4 and mixed with an equal volume of 0.125 mg/mL of butylated oxytoluene in ethanol to precipitate the proteins. Next, each sample was diluted twice with n-hexane. The mixture was vortexed for 5 min for the extraction of vitamins and centrifuged at 3000× *g* for 10 min. The hexane layer was injected onto the HPLC system [35]. Chromatographic separation was carried out on a 0.4–0.6 m 250–300 µL silica gel Analytical Chromatographic Column 6 using a Milichrom 6 chromatograph (Nauchpribor, Russia). A mixture of n-hexane and isopropanol (98.5:1.5, *v*/*v*) was applied as the mobile phase. The eluate was monitored at 292 nm, and the vitamin was identified by the retention time compared with the pure standard (MP Biomedicals, Irvine, CA, USA).

#### 2.4.2. Lipid Composition Analysis

Lipids were extracted with chloroform/methanol (2:1, *v*/*v*) according to Folch et al. [36]. The extracted lipids were spotted onto TLC Silica gel 60 F254 thin-layer chromatography plates (Merck, Darmstadt, Germany) and separated into different fractions of lipid classes using petroleum ether/diethyl ether/acetic acid (90:10:1, *v*/*v*) as the mobile phase. The identification of the fractions was performed using standards such as the phospholipid mixture (P3817, Supelco, St. Louis, USA), cholesterol (C8667, St. Louis, Sigma-Aldrich, USA), glyceryl trioleate (92860, Sigma-Aldrich, St. Louis, USA), and cholesteryl palmitate (C78607, Sigma-Aldrich, St. Louis, USA). The quantitative composition of the fractions was measured at 540 nm for phospholipids, triacylglycerols, and cholesterol esters and at 550 nm for the sterol fraction using an SF-2000 UV/Vis spectrophotometer (Spektr, Saint Petersburg, Russia) [37,38].

The composition of the phospholipid fractions, including phosphatidylinositol (PI), phosphatidylserine (PS), phosphatidylethanolamine (PE), and phosphatidylcholine (PC), was determined by high-performance liquid chromatography using the method of Arduini et al. [39] on a Nucleosil 100-7 column (Elsiko, Moscow, Russia) with the acetonitrile/hexane/methanol/phosphorus acid mixture as a liquid phase (918:30:30:17.5, *v*/*v*) and a UV-spectrophotometer at 206 nm using a Stayer liquid chromatography system (Aquilon, Moscow, Russia). Peaks were identified by referencing the retention times of the authentic standard phospholipid mixture (P3817, Supelco, St. Louis, USA) and phosphatidylserine (P7769, Sigma-Aldrich, St. Louis, USA).

Fatty acid methyl esters (FAME) from the total lipids were prepared using methanol and acetyl chloride. The FAME were separated on an Agilent 7890A gas–liquid chromatograph (Agilent Technologies, Santa Clara, USA) with a flame ionization detector using DB-23 columns (60 m–0.25 mm) (Agilent Technologies, Santa Clara, USA) and nitrogen as the mobile phase. The FAME from fish total lipids were identified by comparison with standard mixes (Supelco, St. Louis, USA). Minor fatty acids constituting less than 1% of the total fatty acids were not identified.

### 2.5. Statistical Analyses

The statistical analyses were performed using R Statistical Software [40]. Differences between samples were estimated using the nonparametric Mann–Whitney U test. Correlations between the investigated parameters were analyzed using Spearman’s rank correlation coefficient with *p*-values adjusted using the Holm method. The weight gain rates in the two groups were compared using an ANCOVA. The differences were considered significant at *p* ≤ 0.05. The values of the studied biochemical parameters are presented as the median values ± half of the interquartile range.

## 3. Results

### 3.1. Fish Survival and Growth

During the observation period, a significant difference in fish mortality was observed between the studied fish groups (Appendix A). The total number of fish deaths was much lower (about 2.5 times) in the group that received dihydroquercetin and arabinogalactan than in the supplement-free group (2.5 vs. 6.5% per season). In the experimental group, the highest mortality rate (0.95% of the total number of fish per month) was observed in August, at the third week since the manifestation of the sporadic bacterial infection at the fish farm. After disease curation, mortality in the supplement-fed group slowed down, with a consistently low rise after October. In the control group that received commercial feed only, the increase in mortality started around the same time (in early August), rising from 0.96% per month to 2% per month by September, and then remained high through to the end of the experiment.

The weight gain in the farmed trout of either group was linear (correlation coef. r = 0.98; Figure 2) and equal in both groups (ANCOVA) during the season. There were no differences between the weight gain rates of the control and experimental groups.

### 3.2. Hepatic Antioxidant Components

The glutathione-mediated antioxidant responses varied within the period of observation, mostly depending on the infectious status of the fish (Figure 3, Appendix A), with maximum glutathione-S-transferase (GST) activity in August. Additionally, a significant positive correlation of the GSH level with the water temperature (r = 0.6; Appendix A) was found. No prominent correlations of fish biometrics with AOS parameters were found. The GSH concentrations in fish tissues revealed no significant diet-related differences in any sampling date except those in November, when an increase (about 1.5-fold) in the hepatic GSH pool was observed in the supplement-free group (Figure 3A). Similarly, the hepatic GST activity in the standard-fed fish was higher (about 2-fold) in November (Figure 3B). The levels of other studied antioxidants, such as hepatic α-tocopherol (vitamin E), and the catalase and peroxidase activity levels were similar in both studied groups throughout the observation period (Appendix A).

### 3.3. Hepatic Lipid and Fatty Acid Profile

In farmed rainbow trout, the fish mass gain was found to be correlated with phosphatidylethanolamine and phosphatidylserine (r = −0.6 both), as well as with sterols and vaccenic 18:1n-7 and palmitoleic 16:1n-7 acids (r = 0.6, 0.7, 0.5, respectively) (Figure 4A), indicating the structural function of these lipids. The water temperature was shown to be negatively correlated with the contents of eicosapentaenoic 20:5n-3 and arachidonic 20:4n-6 fatty acids, the constituents of membrane phospholipids (r = −0.7, −0.6, respectively). In general, the hepatic fatty acid composition was shown to be very similar to that of commercial feed (Figure 4B). Expectedly, abundant components of the fish feed, such as oleic 18:1n-9, linoleic 18:2n-6, alpha-linolenic 18:3n-3, gamma-linolenic 18:3n-6, eicosapentaenoic 20:5n-3, and palmitoleic 16:1n-7 acids, are positively correlated with each other in fish livers, likely due their common exogenous origin (Figure 4A).

In both studied fish groups, the TAGs were intensely accumulated in summer in accordance with the total fat deposition (negative correlation with water temperature, r = −0.5) (Figure 4A and Figure 5) and positively correlated with oleic 18:1n-9 and palmitoleic 16:1n-7 acids (r = 0.5, 0.6, respectively). Although it was unlikely due to the fish feed, greater portions of docosahexaenoic 22:6n-3 and arachidonic 20:4n-6 acids were found in fish livers (Figure 4B). The correlation network analysis indicated the conversion of alpha-linolenic 18:3n-3 to docosahexaenoic acid and linoleic 18:2n-6 to arachidonic acid (r = −0.6 and −0.8, respectively). Similarly, saturated 18:0 fatty acids were transformed to monounsaturated oleic 18:1n-9 and vaccenic 18:1n-7 acids (r = −0.6 and −0.5, respectively) in the livers of the fish (Figure 4A).

The dietary supplementation with arabinogalactan and dihydroquercetin for just a week resulted in an increase in the 15:0 fatty acid and a decrease in the 18:0 fatty acid contents in fish livers (Figure 6A,B). There were no significant intergroup differences in hepatic lipid content while the infection progressed (through August) (Figure 1, Figure 2, Figure 3, Figure 4, Figure 5, Figure 6, Figure 7 and Figure 8, Appendix A). In samples collected on 12 August after antibiotic treatment and the second two-week course of supplement administration, significant decreases in hepatic 18:0 and 16:0 fatty acids were found in fish fed an enriched diet (Figure 6B,C), while decreases in linoleic 18:2n-6 and alpha-linolenic 18:3n-3 were detected in supplement-free trout for the same dates (Figure 7A,B). In the postinfection period, no prominent difference in lipid composition between the fish groups was found; in contrast, by the time the water temperature dropped to 6 °C (November 16), the contents of hepatic sterols and vaccenic acid 18:1n-7 increased and of phosphatidylethanolamine decreased in fish receiving an enriched but not a standard diet (Figure 7C and Figure 8A,B). Similarly, vaccenic acid 18:1n-7 as a component of phospholipids positively correlated with fish weight (Figure 4A). In infected supplement-free fish, the content of vaccenic acid 18:1n-7 decreased while the contents of more easily oxidizable substrates, polyunsaturated docosahexaenoic 22:6n-3 and arachidonic 20:4n-6 acids, did not differ between the groups (Appendix A).

## 4. Discussion

### 4.1. Infections and Mortality

Dihydroquercetin- and arabinogalactan-containing feed had a beneficial effect on fish viability, significantly reducing the cumulative mortality rate within the growing season. The impaired welfare observed in the supplement-free fish group was likely a result of a more severe infection, lower susceptibility to the antibiotic treatment, re-infestation, or other causes. The similar growth rates mean that regardless of the diet, the fish consumed food and effectively converted the obtained energy into body mass. The unperturbed growth but lower viability in fish that were fed a standard diet possibly indicate the sudden death of individuals due to sporadic acute infection rather than the chronic effect of any undetermined hazardous agents on caged fish. We assume that the tested supplement mixture may have a beneficial long-lasting effect on fish viability, probably via increasing the resistance to both infections and re-infections. Our data on the influence of the studied plant-origin substances on the survival and welfare of rainbow trout correspond with data in the literature describing the effects of other related compounds on the welfare and mortality of fish. For example, medicinal plant feed additives enhanced the growth and survival of cultured *Clarias gariepinus* [41]. Dietary medicinal plant extracts improved the growth, immune activity, and survival of tilapia, *Oreochromis mossambicus* [42].

### 4.2. Antioxidant Defense

A number of stress factors (e.g., temperature, bacterial infection, medications) initiating reactive oxygen species (ROS) generation and lipid peroxidation (LPO), and consequently providing oxidative stress, affect the survival and growth of trout [43,44,45,46]. The antioxidant system (AOS) protects an organism from the oxidative stress at the molecular level [43,45]. The AOS includes (1) low-molecular-weight antioxidants (e.g., reduced glutathione, tocopherol, and others) directly interacting with reactive oxygen species and (2) antioxidant enzymes, such as catalase, peroxidase, glutathione-S-transferase, catalyzing the deactivation and elimination of adverse molecules [43,47,48]. The conjugation of pro-oxidants with GSH, either spontaneously or via being catalyzed by glutathione-S-transferase, and the subsequent excretion of the complexes from the organism is one of the main pathways of elimination of ROS and their derivatives [49,50]. Interestingly, dihydroquercetin was reported to directly upregulate glutathione-S-transferases in human cells through the antioxidant-responsive element (ARE) binding the promoter regions of GST genes [51]; however, we did not observe a similar effect in trout. The increased GST activity and GSH content found in fish of the supplement-free group in November suggested enhanced ROS production in their tissues in response to low temperatures. Catalase and peroxidase decompose organic and inorganic peroxides causing membrane LPO and α-tocopherol captures free radicals in cell membranes [43,47,48]. As these antioxidants were unaffected by the infection in both studied fish groups, we can assume that the ROS generation and lipid peroxidation were moderate and did not exceed the antioxidant capacity of the low-molecular-weight antioxidant pool members, such as glutathione. Similarly, dietary dihydroquercetin at a dosage 500 mg (20 times more than what the trout received) produced no effect on hepatic α-tocopherol and peroxidase in broiler chickens [13]. In the supplement-free fish group, glutathione and GST activation in response to ambient temperature but not to the infection is probably sufficient to prevent LPO progression.

### 4.3. Lipid and Fatty Acid Response to Stressors

Being a key component of various metabolic pathways, lipids play a crucial role in the survival and tolerance of an organism to environmental stressors. Cell lipids contain polyunsaturated fatty acids enriched by double bonds within their hydrocarbon chains, which are primary targeted by pro-oxidants and can indicate LPO progression. The main lipid components of the *O. mykiss* liver are membrane phospholipids. Together with sterols (mainly cholesterol), they comprise cell membranes and affect membrane fluidity and the activity of membrane-bound enzymes, ion channels, and receptors [52,53,54]. Membrane phospholipids are also a source of bioactive compounds and messengers [55,56]. Phospholipids include a hydrophilic head, which may include substances such as choline, ethanolamine, or serine and hydrophobic long-chain fatty acids.

The water temperature determines the dissolved oxygen content and feed consumption rate of the fish and eventually their metabolic rate [28], and directly affects the cell membrane stability. Our data on the negative correlation of the eicosapentaenoic 20:5n-3 and arachidonic 20:4n-6 fatty acid contents and ambient water temperature support the observations on their contribution to the maintenance of membrane fluidity under temperature response [52,53,56].

Triacylglycerols (TAGs) are high-energy lipid molecules stored during intense feeding, readily covering the energy costs needed to maintain homeostasis at different life stages or in different environments [54,56]. In both studied fish groups, the TAGs were intensely accumulated in summer in accordance with the amount of supplied feed and total fat deposition, and their content was positively correlated with oleic 18:1n-9 and palmitoleic 16:1n-7 acids, probably indicating their alimentary uptake and deposition as TAG constituents in hepatic fat.

According to previous observations, the fatty acid composition of the fish liver is very similar to that of commercial feed, with an equal ratio of oleic 18:1n-9, linoleic 18:2n-6, alpha-linolenic 18:3n-3, gamma-linolenic 18:3n-6, eicosapentaenoic 20:5n-3, and palmitoleic 16:1n-7 acids, suggesting their common exogenous origin [57,58,59,60]. This is especially true for linoleic 18:2n-6 and alpha-linolenic 18:3n-3 acids, which are not produced by fish and can only be of food origin. Compared to the fish feed composition, greater portions of docosahexaenoic 22:6n-3 and arachidonic 20:4n-6 acids were found in the fish livers, indicating that they are at least partly of endogenous origin. We found supporting evidence of alpha-linolenic 18:3n-3 conversion to docosahexaenoic acid, linoleic 18:2n-6 to arachidonic acid, and saturated 18:0 to monounsaturated oleic 18:1n-9 and vaccenic 18:1n-7 acids, which physiologically occurs in the liver of freshwater fish [59].

The accumulation of 15:0 fatty acid (mostly of bacterial origin) and a decrease in 18:0 coincided with the initial stage of infection, and this response may arise from exposure to stress or pathogens. Although 15:0 acid is known to be of bacterial origin, it still unclear whether it was sourced from pathogenic or symbiotic microbiota or if these components were unequally assimilated from food by the fish in both studied groups [61].

Since linoleic and alpha-linolenic acids are both essential and food-derived, these results can be interpreted as an impaired digestion sign in supplement-free fish. The beneficial effect of dietary arabinogalactan on symbiotic intestinal microflora (bifidobacteria and lactobacilli) as well as the boosting of immunological characteristics were reported previously in mammals [15]. The accelerated postinfection recovery and enhanced assimilation of nutrients observed in fish fed an experimental diet presumably resulted from the antioxidant and immunostimulant activity of the supplement. We can predict diet-related differences in fatty acid utilization pathways in fish; thus, the 18:0 and 16:0 acids could be more readily utilized in monounsaturated acid synthesis in fish fed an enriched diet, whereas linoleic and alpha-linolenic acids could be more intensively used to synthesize polyunsaturated acids in fish receiving a standard diet. The preferred synthesis of unsaturated fatty acids in fish in the control group may have been due to the need to stabilize cell membranes in animals more perturbed by the infection, because in response to this pathogen cells produce ROS both possessing antimicrobial activity and initiating LPO [43,44,45]. A possible explanation for the vaccenic acid 18:1n-7 decrease detected in the supplement-free fish is its oxidation by ROS. However, the contents of polyunsaturated docosahexaenoic 22:6n-3 and arachidonic 20:4n-6 acids, which are more easily oxidizable substrates in cell [56,59], did not differ between the groups, excluding LPO progression and severe oxidative stress.

Sterols (mainly consisting of cholesterol) and phosphatidylethanolamine both comprise important indexes of membrane fluidity, such as the ratios of phospholipids to cholesterol and phosphatidylcholine to phosphatidylethanolamine [62]. The regulation of membrane fluidity is essential for adaptation to temperature variations; therefore, it can be assumed that fish from the control group having low cholesterol and increased phosphatidylethanolamine may be less adapted to wintering. Cell membrane stabilization and the regulation of membrane permeability were proposed earlier [63] as mechanisms of action of the plant flavonoid extract Legalon^®^ (including dihydroquercetin), a licensed drug against liver cirrhosis [64]. It should also be noted that a depressing effect of dihydroquercetin on liver cholesterol through the inhibition of its hepatic biosynthesis and secretion of the cholesterol carrier apolipoprotein (apoB) was reported previously for human cells [65,66]. However, this effect was not observed during our study on trout, and the opposite effect was observed in the November samples.

Due to the poor response of the antioxidant system and moderate loss of lipid substrates for oxidation during the bacterial infection, the protective effect of dihydroquercetin and arabinogalactan against severe oxidative stress [43,44,45] in fish fed supplements remains unproven. However, certain differences in cell membrane organization and antioxidant response induced by low ambient temperatures and associated food shortages were revealed between both fish groups.

## 5. Conclusions

Some beneficial effects of the dietary mix of dihydroquercetin and arabinogalactan on physiology responses of farmed rainbow trout, *O. mykiss,* were revealed. The welfare of the fish in both groups was impacted by a sporadic bacterial infection, including an increased mortality rate and impaired assimilation of nutrients, particularly the malabsorption of essential linoleic 18:2n-6 and alpha-linolenic 18:3n-3 fatty acids. The feed supplementation partially reduced both the lethality of the fish stock and the impairment of their digestion during the infection and at the postinfection period. Therefore, the enhanced resistance to the infection and an accelerated postinfection recovery rate observed in fish fed a supplemented diet could be the results of the biological activity of plant substances. Presumably, plant-origin substances stimulate vitality and natural resistance in farmed fish though increasing the efficacy of fish production. Although the benefits of these feed supplements in fish production are indisputable, further experiments under controlled conditions are required to study the mechanism of the immunomodulating activity of the dietary mix in detail.

## Figures and Tables

**Figure 1 animals-13-01345-f001:**
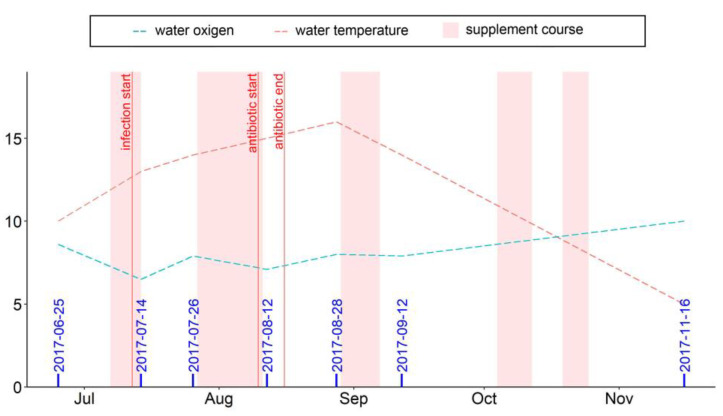
Dates of sampling (blue digits), water temperature (°C), and dissolved oxygen content (mg/L) during the observation period.

**Figure 2 animals-13-01345-f002:**
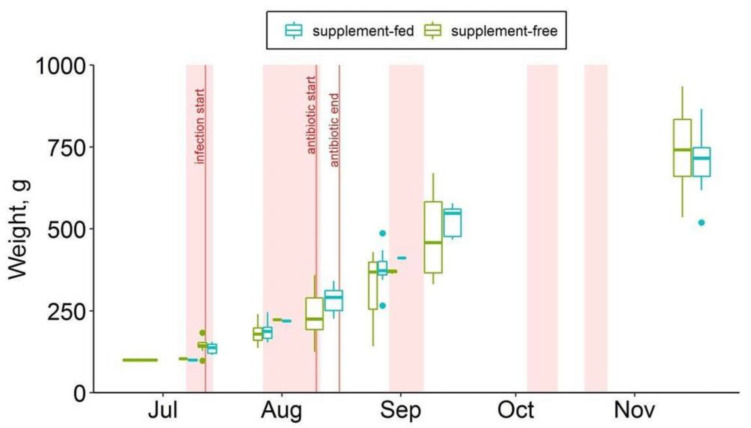
Body weights of supplement-free and supplement-fed *O. mykiss* fish over the observation period.

**Figure 3 animals-13-01345-f003:**
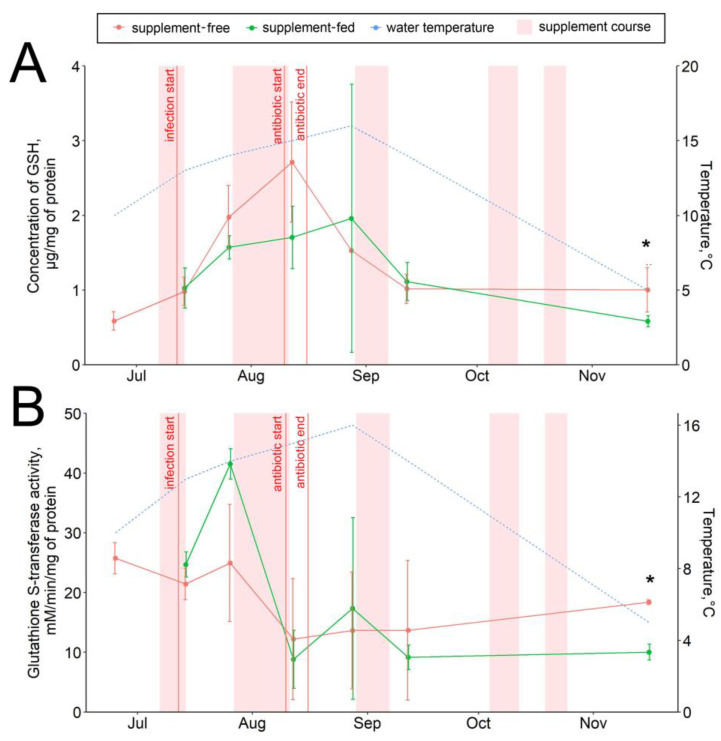
Hepatic antioxidants in supplement-free and supplement-fed *O. mykiss* fish: (**A**) reduced glutathione content; (**B**) glutathione-S-transferase activity. * Statistically significant differences between the diets in the same sampling date.

**Figure 4 animals-13-01345-f004:**
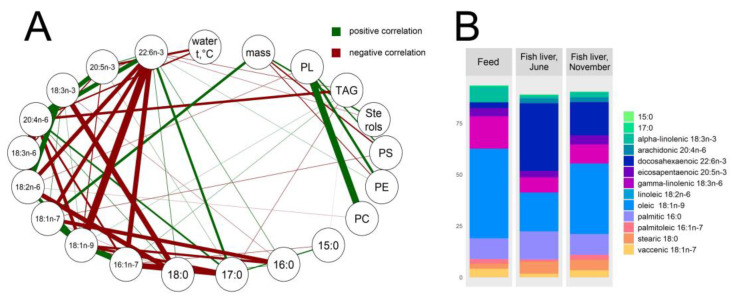
Lipid landscapes in *O. mykiss* liver samples and the commercial feed used. (**A**) Network of prominent correlations (r ≥ 0.4, *p* ≤ 0.05) of studied lipid components, water temperature, and fish weight. Abbreviations: PL, phospholipids; PC, phosphatidylcholine; PE, phosphatidylethanolamine; PS, phosphatidylserine; TAG, triacylglycerols. A thicker line means a stronger correlation. (**B**) Fatty acid ratios (% of total fatty acids) in fish livers and feed pellets.

**Figure 5 animals-13-01345-f005:**
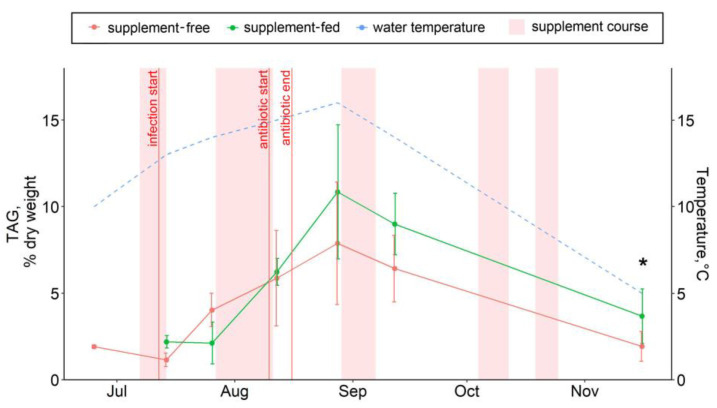
The hepatic triacylglycerol contents in supplement-free and supplement-fed *O. mykiss* fish. * Statistically significant differences between the diets on the same sampling date.

**Figure 6 animals-13-01345-f006:**
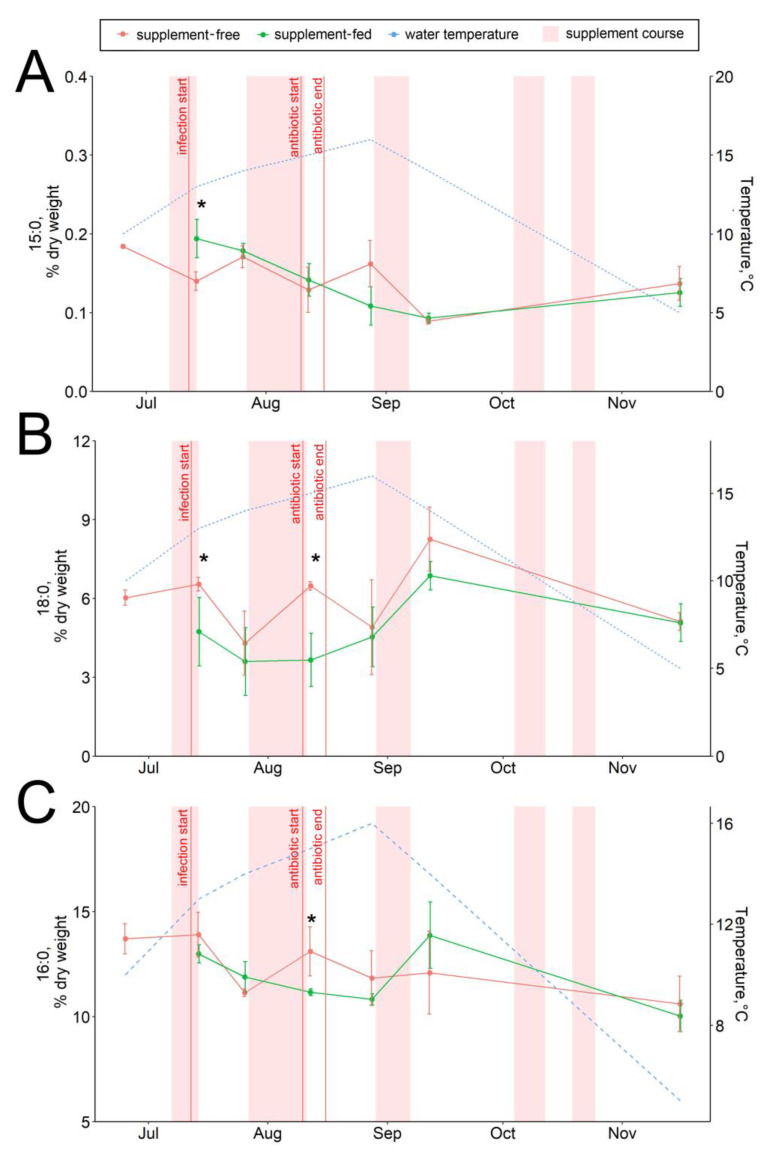
Hepatic saturated fatty acid contents in supplement-free and supplement-fed *O. mykiss* fish: (**A**) 15:0; (**B**) 18:0; (**C**) 16:0 fatty acids. * Statistically significant differences between the diets on the same sampling date.

**Figure 7 animals-13-01345-f007:**
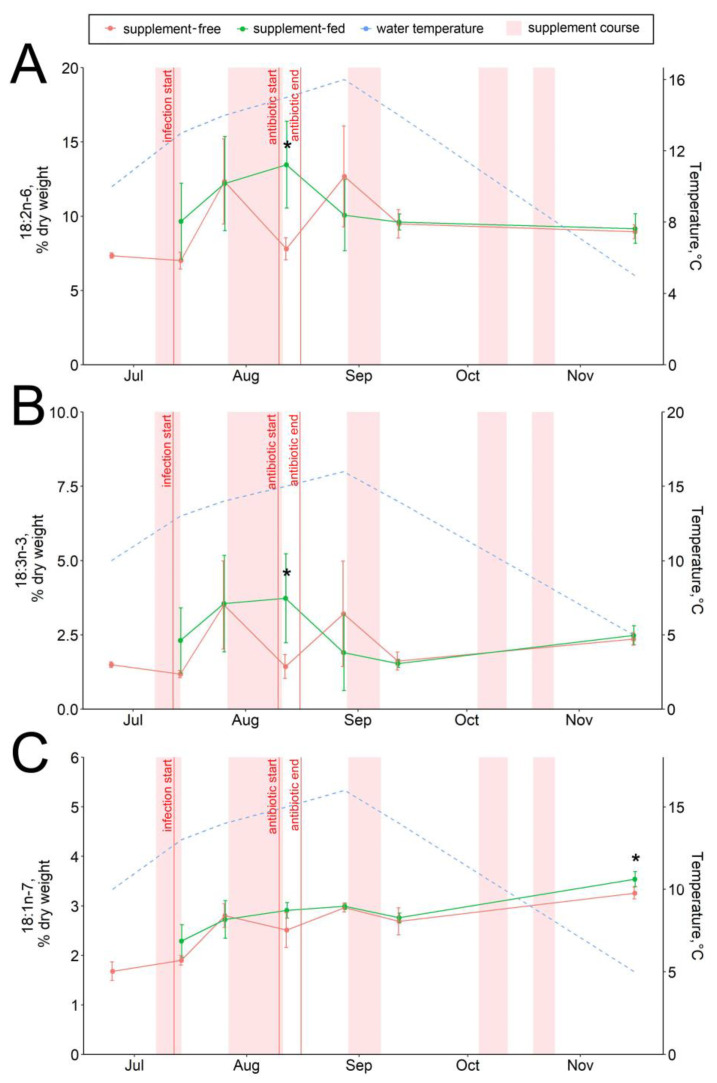
Hepatic unsaturated fatty acid contents in supplement-free and supplement-fed *O. mykiss* fish: (**A**) linoleic 18:2n-6; (**B**) alpha-linolenic 18:3n-3; (**C**) vaccenic 18:1n-7. * Statistically significant differences between the diets on the same sampling date.

**Figure 8 animals-13-01345-f008:**
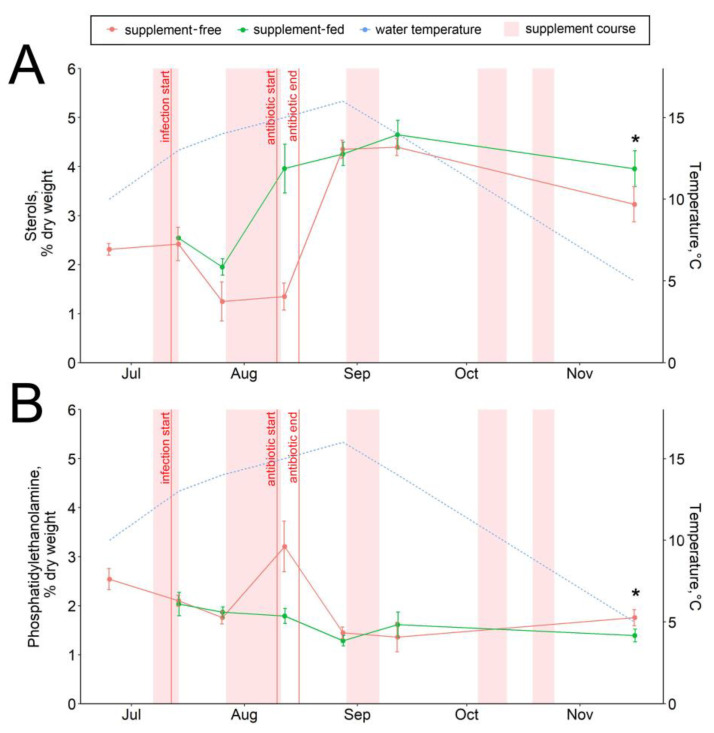
Hepatic sterol (**A**) and phosphatidylethanolamine (**B**) contents in supplement-free and supplement-fed *O. mykiss* fish. * Statistically significant differences between the diets on the same sampling date.

## Data Availability

All data are presented in the paper.

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
