# Peer review of "Survival, Growth Performance, and Hepatic Antioxidant and Lipid Profiles in Infected Rainbow Trout (Oncorhynchus mykiss) Fed a Diet Supplemented with Dihydroquercetin and Arabinogalactan"

_animals, 2023, doi:10.3390/ani13081345_

Round 1
Reviewer 1 Report
Manuscript Number: animals-2263449-peer-review-v1
Reviewer Recommendation and Comments:
This manuscript is important and innovative research. I suggest minor revisions:
Lines 45 to 49: It needs to be revised.
Line 57: Similar studies should be mentioned in the introduction.
Line 94 to 96: Remove unnecessary content
Line 103: Determine the specifications of the device for measuring water factors
Line 121: What dose of clove oil was used?
Line 141: be summarized
Line 224: What does it mean (median ± half)
Line 227: It needs to be revised
Line 327: Comparison with other research
Line 339 and 365: Explain the scientific reasons for the changes
Line 442: be summarized
Author Response
Authors are deeply grateful to the reviewer for careful reading of the manuscript, valuable comments, and suggestions. We agree with the majority of the comments, and the revisions made are explained below.
Lines 45 to 49: It needs to be revised.
The sentences have been revised to clarify main findings and conclusions.
Dietary supplementation with studied plant-origin substances, such as dihydroquercetin and arabinogalactan, decreases lethality in the fish stock presumably though stimulation of a natural resistance in farmed fish thus increasing economic efficacy of fish production. From the sustainable aquaculture perspective, natural additives also diminish anthropogenic transformation of aquaculture-bearing water bodies and their ecosystems.
Line 57: Similar studies should be mentioned in the introduction.
Relevant citations have been added.
Line 94 to 96: Remove unnecessary content
Unnecessary content has been removed
Line 103: Determine the specifications of the device for measuring water factors.
The missing information has been added.
The water temperature and dissolved oxygen, measured daily with S9 Seven2Go pro (Mettler Toledo, Switzerland) throughout the study period, were between 6 and 18°С and 5 and 11 mg/L, respectively (Figure 1).
Line 121: What dose of clove oil was used?
The missing information has been added.
At indicated sampling dates, eight fish from each cage were euthanized with an overdose of clove oil (250 mg L⁻¹).
Line 141: be summarized
Some information has been added, this section has been summarized.
Line 224: What does it mean (median ± half)
The values of studied biochemical parameters are presented as the median ± half of the interquartile range. Interquartile range is a measure of statistical dispersion, which is the spread of the data. Half of the interquartile range or the semi-interquartile range is half of the difference between the third quartile (Q3) and the first quartile (Q1).
Line 227: It needs to be revised
Some information has been added
Line 327: Comparison with other research
Comparison with other research has been added.
Line 339 and 365: Explain the scientific reasons for the changes
Some explanations were added to the text.
Line 442: be summarized
Conclusion section has been rewritten to summarize our findings on the supplement effects and further perspectives.
Reviewer 2 Report
The work presented to me for review is very interesting and fits into the trend of non-invasive care for the welfare of farmed fish. Some parts of the manuscript, however, are not clear described, such as the mortality reported per month and for the entire duration of the experiment. These data appear in several places in the manuscript, which causes informational dissonance.
1. Suggestion of specifying citations 8, 9 and 10, in lines 65 - 67, in the form of indicating the specific subject of research of the cited publications.
2. The need to specify in a separate paragraph at the end of the "Introduction" part the purpose of the work, starting with the words: The aim of this study was ... The text from line 75 to line 81 after grammatical correction can be used here.
3. Line 78. Was the episode of bacterial infection intended and deliberate by the authors, or was it due to natural causes? The further text of the publication shows that it was natural, but it must be clearly articulated at the very beginning of the manuscript.
4. Suggestion to remove the information on the total number of fish used in the experiment in the "Materials and Methods" section (line 85) and supplement this section with data on the total biomass of fish in each cage.
5. The need to literally indicate in the "Materials and Methods" part the number of repetitions in each variant of the experiment, as well as the number of variants.
6. The need to supplement the "Materials and Methods" part, for example in line 103 or 104, with data on the device used to measure basic water parameters (company, type of device).
7. Suggestion to indicate in the "Materials and Methods" section, point 2.2, used for fish euthanasia, a dose of clove oil.
8. Was the amount of food rations calculated daily with the correction of the current biomass adequate to the level of fish mortality, or was it calculated once a month? Section 2.3.
9. Suggestion to move the content of point 4.1 in the "Discussion" section to the "Results" section.
10. Suggestion to delete the first sentence (lines 376-377) due to an obvious physical fact.
11. The need to check the consistency of the reference cited in the text with its list at the end of the manuscript.
Author Response
Authors are deeply grateful to the reviewer for careful reading of the manuscript, for valuable comments and suggestions. We agree with the majority of the comments, explanation for them is given below.
- Suggestion of specifying citations 8, 9 and 10, in lines 65 - 67, in the form of indicating the specific subject of research of the cited publications.
The missing information has been added. New version of the sentences:
There have also been attempts to use dietary dihydroquercetin in veterinary medicine, which had either beneficial effect –increasing the immune status of gilthead seabream [11] and suppressing of Cd toxicity in zebrafish embryos [12] or none – no impact on growth performance or any of the studied physiological variables [13].
- The need to specify in a separate paragraph at the end of the "Introduction" part the purpose of the work, starting with the words: The aim of this study was ... The text from line 75 to line 81 after grammatical correction can be used here.
The paragraph has been changed.
The aim of the study was to estimate some physiological and biochemical indices in farmed rainbow trout (Oncorhynchus mykiss) received either a standard diet or a diet supplemented with dihydroquercetin and arabinogalactan throughout a growing season, for five months. Due to an episode of a natural bacterial infection occurred during the observation period, the antioxidant and immunomodulating properties of a natural additive could able to be verified.
- Line 78. Was the episode of bacterial infection intended and deliberate by the authors, or was it due to natural causes? The further text of the publication shows that it was natural, but it must be clearly articulated at the very beginning of the manuscript.
The missing information has been added. New version of the sentence:
Due to an episode of a natural bacterial infection occurred during the observation period, the antioxidant and immunomodulating properties of a natural additive could able to be verified.
- Suggestion to remove the information on the total number of fish used in the experiment in the "Materials and Methods" section (line 85) and supplement this section with data on the total biomass of fish in each cage.
The missing information has been added. New version of the sentences:
Healthy rainbow trout Oncorhynchus mykiss yearlings (average weight 100.1±2.3 g, age 1+) were placed in four cages with initial stocking density of 2.1 kg/m3; the total biomass of fish in each cage was around 886 kg.
- The need to literally indicate in the "Materials and Methods" part the number of repetitions in each variant of the experiment, as well as the number of variants.
We suppose that all information about the number of repetitions is in the text.
Each dietary was experimented in duplicate.
Rainbow trout Oncorhynchus mykiss yearlings (average weight 100.1±2.3 g, age 1+) were placed in four cages. Fish were fed one of two diets (in duplicate), either the commercial diet BioMar without any supplements (control diet) or the same commercial diet supplemented with 25 mg/kg dihydroquercetin and 50 mg/kg arabinogalactan.
- The need to supplement the "Materials and Methods" part, for example in line 103 or 104, with data on the device used to measure basic water parameters (company, type of device).
The water temperature and dissolved oxygen, measured daily with S9 Seven2Go pro (Mettler Toledo, Switzerland) throughout the study period, were between 6 and 18°С and 5 and 11 mg/L, respectively (Figure 1).
- Suggestion to indicate in the "Materials and Methods" section, point 2.2, used for fish euthanasia, a dose of clove oil.
The missing information has been added.
At indicated sampling dates, eight fish from each cage were euthanized with an overdose of clove oil (250 mg L⁻¹).
- Was the amount of food rations calculated daily with the correction of the current biomass adequate to the level of fish mortality, or was it calculated once a month? Section 2.3.
The amount of food rations was calculated daily according to the protocol considering individual fish mass, current biomass in each cage, water temperature and oxygen.
The missing information has been added to the section 2.1. Fish maintenance and feeding.
- Suggestion to move the content of point 4.1 in the "Discussion" section to the "Results" section.
Comparison with other research has been added. We prefer not to move the content of point 4.1 in the "Discussion" section to the "Results" section.
- Suggestion to delete the first sentence (lines 376-377) due to an obvious physical fact.
We prefer not to delete this sentence because this sentence is essential for discussion
- The need to check the consistency of the reference cited in the text with its list at the end of the manuscript.
The consistency of the reference cited in the text with its list at the end of the manuscript has been checked.
Round 2
Reviewer 2 Report
After the correction and the authors' response to my comments and suggestions, I have no objections to the revised version of the manuscript. Thus, I suggest admitting the work to further stages of the publication process.